# A Psychoacoustic Approach to Building Knowledge about Human Response to Noise of Unmanned Aerial Vehicles

**DOI:** 10.3390/ijerph18020682

**Published:** 2021-01-14

**Authors:** Antonio J. Torija, Charlotte Clark

**Affiliations:** 1Acoustics Research Centre, University of Salford, Manchester M5 4WT, UK; 2Ove Arup & Partners, Acoustics, 13 Fitzroy Street, London W1T 6BQ, UK; Charlotte.Clark@arup.com

**Keywords:** drone noise, community noise impact, health effects of noise

## Abstract

We are on the cusp of a revolution in the aviation sector, driven by the significant progress in electric power and battery technologies, and autonomous systems. Several industry leaders and governmental agencies are currently investigating the use of Unmanned Aerial Vehicles (UAVs), or “drones” as commonly known, for an ever-growing number of applications—from blue light services to parcel delivery and urban mobility. Undoubtedly, the operation of UAVs will lead to noise exposure, which has the potential to become a significant public health issue. This paper first describes the main acoustic and operational characteristics of UAVs, as an unconventional noise source compared to conventional civil aircraft. Gaps in the literature and the regulations on the noise metrics and acceptable noise levels are identified and discussed. The state-of-the-art evidence on human response to aircraft and other environmental noise sources is reviewed and its application for UAVs discussed. A methodological framework is proposed for building psychoacoustic knowledge, to inform systems and operations development to limit the noise impact on communities.

## 1. Introduction

The long-term strategy to decarbonise transport is driving a revolution in the aviation sector. A number of electric and autonomous aerial technologies are currently under development, from Unmanned Aerial Vehicles (UAVs), commonly called “drones”, to regional/short haul hybrid electric aircraft. These novel aerial technologies can be classified as (1) UAVs, varying in size (from a few grams to hundred kilograms) and type (e.g., fixed wing vs. multi-rotors), with unlimited applications from parcel delivery to survey and search and rescue; (2) Urban Air Mobility, to transport people within cities—these vehicles are expected to be based on electric Vertical Takeoff and Landing (eVTOL) technology; and (3) electric/hybrid electric aircraft for regional to short-haul routes. This paper focused on UAVs, as the aerial technology most likely to be widely adopted. Notwithstanding this, the findings and discussion of this paper could be applicable to all three categories above.

Schäfer et al. [1] discussed how a transition to all-electric aircraft might significantly reduce the environmental impacts of aviation, e.g., eliminate direct combustion emissions and thus remove associated direct CO_2_ and non-CO_2_ warming. According to these authors, the adoption of all-electric, narrow-body aircraft will not take place without a significant advance in battery technology. However, battery-operated UAVs have been proposed as a sustainable aerial transportation system able to significantly reduce the carbon footprint in cargo transport and parcel delivery [2,3]. Elsayed and Mohamed [4] found that parcel delivery using UAVs produces lower greenhouse gas (GHG) emissions than ground delivery per parcel-km (although compared to electric vehicles, UAVs showed only slightly lower GHG emissions for deliveries in urban areas and 30% more GHG emissions in rural areas). UAVs could also bring important societal benefits, such as urgent medical deliveries [5].

The expansion of the UAV sector requires building knowledge and also overcoming some important challenges for their integration into the existing aviation airspace, infrastructure and aviation systems. Some of these challenges are (1) regulation; (2) vehicle testing and certification; (3) operational safety; (4) vehicle-to-vehicle communication; (5) cybersecurity; (6) energy sources; (7) airspace management; (8) new infrastructure; (9) developing new business markets; (10) standardisation of noise assessment; (11) human response to noise and visual pollution; and (12) public acceptance [6].

The noise generated by UAV operations will certainly alter existing soundscapes [7]. UAVs generate an unconventional noise signature, which does not qualitatively resemble conventional aircraft noise [8]. This is a source of noise that people have not before encountered, and that will likely impact communities not currently exposed to aircraft noise. For these reasons, it is widely accepted that noise is one of the main limiting factors for public acceptance and adoption of UAVs. It is unquestionable that, if not addressed appropriately, noise produced by UAVs could lead to important tension with exposed communities, therefore putting at risk the significant environmental and societal benefits UAVs might bring.

The objectives of this paper are to describe the main factors that should be considered to assess the human response to UAV noise, and review and discuss whether current regulation, noise metrics and evidence of health effects of aircraft noise could be of application to UAV noise. Furthermore, this paper describes a psychoacoustic approach to build knowledge about human response to UAV noise, to inform the design and operation of UAVs for reducing their impact on quality of life and health.

## 2. Drone Noise

UAVs are expected to be quieter than conventional aircraft. In a recent study carried out by Read et al. [9], the A-weighted maximum noise level normalized to a distance of 400 ft (L_A,max_ at 400 ft) for four small- to medium-size UAVs ranges from 50.1 dB(A) to 64.1 dB(A) (mean = 55.1 dB(A), standard deviation = 6.2 dB(A)). These four vehicles are the Skywalker X-8 (fixed wing of 3 Kg), DJI M200 (quadcopter of 6.1 Kg), Yuneed Typhoon (hexcopter of 2.4 Kg) and GD28X (octocopter of 20.4 Kg); and the operation measured was a fast, level flyover over the microphone. According to the Aircraft Noise and Performance (ANP) database [10], at 400 ft, the average L_A,max_ of a conventional Airbus A320 (CFM56-5A1 engine) and Boeing 737-8MAX (CFM Leap1B-27 engine) is 95.6 dB(A) (SD = 2.9 dB(A)) and 90.1 dB(A) (SD = 3.1 dB(A)), respectively. Note that these average values of L_A,max_ are for takeoff operations with a power setting ranging from 12,000 lbf to 22,500 lbf (Airbus A320), and from 10,000 lbf to 24,500 lbf (Boeing 737-8MAX).

Aircraft only fly at 400 ft height above the ground at distances closed to the boundaries of an airport. Focusing on takeoff operations, a given aircraft will accelerate from the Start-of-Roll (SOR) point and climb to a cruise altitude (usually about 30,000 ft). The flight (vertical) flight profile will depend on the specific aircraft and Maximum Takeoff Mass (MTOM). In order to assess the possible range of aircraft noise levels at communities living around airports, the L_A,max_ values at two distances (i.e., 10 km and 25 km) from the SOR point are shown for both an Airbus A320 and Boeing 737-8MAX in Table 1.

However, compared to conventional aircraft, UAVs will operate much closer to the ground. For instance, the “open category” of UAVs are limited to operations in the visual line of sight (VLOS), below 120 m (or 400 ft) altitude [12]. Figure 1 shows the differences in frequency spectra between two UAVs (DJI M200 and Yuneec Typhoon [9]) and two conventional civil aircraft (Boeing 737 MAX 8 and Airbus A320). Note that the frequency spectra were normalised to 65 dB(A) to ease the visualisation of such differences. The Boeing 737-8MAX and Airbus A320 aircraft were recorded at a location approximately 900 m from the end of the south runway of Heathrow airport (approx. 4.5 km from the SOR point), during takeoff operations. The height of the aircraft passing over the measurement point was estimated as 435.2 ± 57.4 m [13]. The DJI Matrice 200 and Yuneec Typhoon UAVs were recorded during level flyover at a 150 ft (i.e., 45.7 m) nominal height above the microphone.

As shown in Figure 1, small to medium UAVs are characterised by significant sound radiation at high frequencies. For conventional fixed-wing aircraft, a significant attenuation of high frequencies is observed due to atmospheric absorption in the long-range propagation. As stated by Gwak et al. [14], one of the critical differences between the sound generated by UAVs and conventional civil aircraft is the concentration of acoustic energy in the high frequency region. This, added to the reduced effect of atmospheric absorption due to the closed proximity between the vehicle and receiver, seems to indicate that high frequency noise is likely to be an important factor for noise annoyance due to UAV operations [15].

The high frequency noise content of UAVs can be observed in Figure 2. As seen in this figure, both UAVs (i.e., DJI M200 and Yuneec Typhoon) radiate significant high frequency noise (especially between 2 kHz and 4 kHz). This high frequency noise can be explained by blade self-noise phenomena, including turbulent boundary layer trailing edge noise [16], interactions between adjacent rotors [15] and the electric motors due to force pulses as the magnets and armature interact [17].

The sound produced by UAVs is mainly tonal in character, with the multiple complex tones at the harmonics of the blade passage frequency (BPF) of each rotor [14,15,16,17,18]. This can be observed in Figure 1 and Figure 3 for both of the UAVs considered in this paper. In Figure 3, for the DJI M200 UAV (quadcopter), two spectral lines corresponding to the BPFs of the two sets of rotors are observed at 120 and 140 Hz. For the Yuneec Typhoon UAV (hexcopter), spectral lines corresponding to the rotors’ BPFs are clustered at about 200 Hz. Harmonics at these BPFs are also observed in Figure 3 (more clearly for the Yuneec Typhoon UAV).

As described by Alexander et al. [19], ambient weather conditions influence the level, frequency composition and temporal characteristics of the sound produced by UAVs. In order to compensate for wind gusts, UAV rotors vary their rotational speed to maintain vehicle stability [15,16]. This leads to an unsteady acoustic signature with rapid temporal fluctuations of the tonal components. The changes in rotor rotational speeds, due to minor variations in the blade and motor properties, and offsets to the UAV centre of gravity also lead to a decrease in the maximum amplitudes of the tonal components and a dispersion of the spectral lines at higher harmonics of the BPFs [15,17].

For all these reasons, authors like Christian and Cabell [8] have suggested that the sound produced by drones does not resemble qualitatively the sound generated by contemporary civil aircraft. Torija and Li [18] have found that even with the same L_Aeq_ (65 dBA), the preference of the sounds of a small quadcopter was 33% lower than the preference of sound of conventional civil aircraft.

## 3. Regulation and Metrics

The current regulation for noise certification of civil aviation seems to be inadequate for capturing the noise features (and cover operational characteristics) of UAVs [20]. Senzing and Marsan [20] discussed the main limitations of the noise certification methods for fixed-wind and rotary-wing aircraft to be applied for UAV vehicles. For instance, certification methods for conventional fixed-wing aircraft assume primary noise effects close to an airport, while for UAVs the noise during en-route operations might be a significant issue due to the closer proximity of UAVs to people.

To address this issue, the European Commission has issued the Implementing Regulation 2019/945 [21]. This regulation is only of application for the “open” category, i.e., a UAV with an MTOM lower than 25 kg. The Annex Part 15 of this regulation presents the maximum A-weighted sound power level (L_wA,max_) per class of UAV (note that L_wA,max_ values are also provided for classes C1 and C2 [12]). The measurement procedure is based on the ISO 3744:2010 [22], and therefore requests the determination (in anechoic chambers) of the sound power levels under free field conditions over a reflecting plane. This test method has been criticised as insufficient and non-optimal to account for the noise produced by UAVs during actual flight operations. Wieland et al. [23] discuss in detail the context for the development of the method in Regulation 2019/945, and outline the main findings and discussions about this noise test code within the ASD-STAN/D5/WG8 working group on Unmanned Aircraft Systems. The noise produced by a UAV during hover has important differences with the noise produced by the same UAV during other manoeuvres like takeoff, landing or flyover, particularly in relation to directivity and prominence of the tonal components [16]. Therefore, the measurements carried out according to Regulation 2019/945 might not reflect the actual noise of the UAVs during other manoeuvres. Furthermore, the test code is based on the A-weighted sound power level, and this noise metric is unable to account for some of the main characteristics of UAV noise (e.g., tonal components, rapid fluctuations in the sound levels, high frequency noise, etc.). In the US, the American Standards Committee (ASC) S12 Working Group 58 (WG58) has agreed not to use the test code described in Regulation 2019/945, and is currently discussing the certification methods for small to medium UAVs [24]. As the test code in Regulation 2019/945, the WG58 is also considering noise testing of UAVs in anechoic chambers. However, this presents important issues: (1) the effects of flow recirculation on the acoustic measurements of UAVs in closed anechoic chambers [25]; and (2) the effect of weather conditions on the sound radiated by UAVs [19].

Depending on the specific certification procedure, a different metric is used. Aircraft noise metrics for certification include the broadband frequency-weighted noise levels, such as the maximum A-weighted sound level (L_A,max_) and the A-weighted Sound Exposure Level (L_AE_). The Sound Exposure Level is numerically equivalent to the total sound energy of an aircraft event. However, these two metrics do not account for certain features, such as the presence of tonal noise. The Effective Perceived Noise Level (EPNL) is the primary metric for the noise certification of fixed-wing and rotary-wing aircraft [26]. The EPNL accounts for the calculation of the Perceived Noise Level [27], which is an indicator of the overall perceived loudness, and is based on the Noy scale (i.e., a scale derived from a combination of amplitude and frequency). The EPNL also includes a correction to account for the duration of the exposure, and a tonal penalty based on the level of the strongest protruding tone [13]. The EPNL is calculated according to a procedure developed by the Federal Aviation Administration (FAA) [28].

These metrics have important limitations to appropriately assess the human response to UAV noise. To start with, these metrics assume a relatively constant noise source [20]. However, as discussed in Section 2, the effect of weather conditions and the control system to maintain vehicle stability are likely to cause rapid fluctuations of the sound levels over time. Therefore, the perception of these rapid sound fluctuations is unlikely to be captured by any of the current noise certification metrics. Moreover, UAV noise is likely to present multiple complex tones at harmonics for the BPFs of each of the multiple rotors operating. The EPNL is the only noise certification metric accounting for a tonal correction, but it seems to be unable to account for the perceptual effects of multiple tonal components distributed along the frequency spectra [13]. Another limitation of the EPNL is that its calculation only considers the one-third octave bands, with nominal centre frequencies between 50 Hz and 10 kHz. However, there is an important high frequency noise content (between 10 kHz to 20 kHz) in the UAV noise signatures [16].

Read and Roof [26] suggest that metrics optimised for UAV noise should include a finer resolution in both time and frequency and the ability to account for the presence of multiple complex tones. Sound quality metrics (e.g., loudness, sharpness, tonality, roughness, fluctuation strength and impulsiveness) are able to accurately assess how the human auditory system reacts to different acoustic features [29]. Loudness is a measure of the sensation of the sound intensity and sharpness measures the sensation of high frequency noise. Fluctuation strength and roughness describe the perception of slow and rapid temporal fluctuation of the sound, respectively. Impulsiveness accounts for the perception of sudden changes in the sound level. Tonality is a metric aimed at identifying and quantifying the perceptual effects of tonal components in a given sound. Sharpness, tonality and fluctuation strength have been found well correlated with rotorcraft noise annoyance [30,31]. In a study investigating the noise annoyance of hovering UAVs, Gwak et al. [14] found that the noise annoyance of UAVs (with MTOM ranging from 113.5 g to 11 kg) was highly related to loudness, sharpness and fluctuation strength. Torija and Li [18] investigated the preference of sounds generated by a single quadcopter (DJI Phantom 3) during level flyover operations at different distances and with different payloads (to vary the operating power of the vehicle). These authors implemented a series of multilevel linear models with the reported preference as the dependent variable. In these models, the intercept was fixed for all participants, and a series of sound quality metrics (and interactions between them—e.g., loudness–tonality interaction) were set to vary randomly across participants. These authors found that a multilevel linear model with a fixed intercept, a tonality metric, and the interaction between the loudness and sharpness metrics as random effects, estimated the reported preference for the quadcopter sounds to be R^2^ = 0.69.

In research on the effects of a hovering UAV on the perception of several urban soundscapes, Torija et al. [7] suggested that the annoyance reported for the soundscapes with the UAV hovering was highly influenced by the specific noise features of the UAV. They also discussed that the reported annoyance seemed to be also influenced by non-acoustic factors, such as the visual scene and expectation for the particular soundscape. Christian and Cabell [8] discussed that noise metrics aimed to provide an accurate assessment of the UAV noise might require the addition of corrections to account for the particular characteristics of their operations (e.g., a loitering penalty to account for the time of exposure). From the findings of Christian and Cabell [8], it might be concluded that the qualitative characteristics of UAV noise are important factors to understand the intrinsic annoyance of these vehicles, and therefore should be considered for the development of noise metrics and to identify acceptable noise levels.

With the deployment of UAVs at scale, their operation will lead to a series of bypassing events that are likely to be highly noticeable (due to their significant content in high frequency and close distance to communities). This intermittent noise, with sequences of noise events emerging above the existing background noise, are likely to cause higher attention to the noise source, and lead to an increase in annoyance [32]. This “eventfulness” may be described by noise metrics such as the Intermittency Ratio (IR). This metric, developed by Wunderli et al. [33], accounts for “the proportion of the acoustical energy contribution in the total energetic dose that is created by individual noise events above a certain threshold”. Emerging or salient noise events can divert attention from the task at hand to the noise source, and therefore reduce task performance [34]. Moreover, it is well accepted that noise events have to be noticed in order for them to contribute to an overall impression of annoyance [35,36]. Therefore, notice-event models (e.g., [36]) might aid in the assessment of soundscapes dominated by bypassing UAV noise events. Furthermore, metrics accounting for the L_A,max_ and number of events (e.g., Noise and Number Index—NNI [37]) should also be considered for predicting the noise annoyance caused by UAV operations.

## 4. What Can We Learn from Civil Aviation (and Other Environmental Noise Sources) about Human Response to Aircraft Noise

### 4.1. Overview

At present, very little is known about how communities may respond to UAVs when operating at scale. As outlined in Section 2, significant research has been carried out to understand the main acoustic properties of UAVs. However, UAV noise has yet to be assessed in terms of human and community response. The operation of UAVs will lead to communities, both in urban and rural settings, becoming newly exposed to noise and also exposed to a new unfamiliar noise signature.

Development of a research methodology can be informed by consideration of the available evidence in relation to environmental noise effects on health, wellbeing and quality of life, for several reasons. Firstly, it has been common practice within the field of noise and health, as the evidence for its effects has built up over the past decades, to apply knowledge obtained about one source of environmental noise to other environmental sources. For example, the United Kingdom government’s methodology for monetising the effects of environmental noise on health applies evidence for an exposure–response function (ERF) between road traffic noise to aircraft noise and railway noise, where evidence for those sources is lacking [38]. Secondly, whilst there are small differences in thresholds for effects, for example aircraft noise is more annoying than road traffic noise and road traffic noise is more annoying than railway noise at the same sound level [39], these differences do not detract from establishing effects per se. Thirdly, both aircraft and drones are aerial vehicles, so they may, despite the differences already described in their noise signatures, have similar effects on health and wellbeing. This leads to the question of what we can learn from studies of environmental noise, about human response to noise, including aircraft noise, road traffic noise and railway noise, with the objective to inform the knowledge needed to build these new aerial vehicles.

### 4.2. Community Noise Annoyance

Annoyance is the predominant community response in a population exposed to aircraft noise. Annoyance describes negative reactions to noise, such as disturbance, irritation, dissatisfaction and nuisance [40], as well as an emotional/attitudinal response [41]. Annoyance is commonly used to measure the impact on quality of life of environmental noise exposure on communities around airports and to derive ERFs, which indicate the percentage of the population highly annoyed (%HA) by a given noise source in the community of interest [42].

Recent years have seen an increase in the strength of the evidence linking aircraft noise to health [43,44]. Long-term exposure to aircraft noise is linked to a range of health outcomes, including sleep disturbance [45], increased cardiovascular and metabolic ill-health [46,47], depression and anxiety [48,49] and poorer academic performance in children [50,51]. The recent WHO Environmental Noise Guidelines for the European Region propose that exposure to aircraft noise above 45 dB L_den_ is associated with adverse health effects and exposure above 40 dB L_night_ is associated with adverse effects on sleep [52]. The WHO Guidelines for road traffic noise and railway noise are both set at slightly higher exposure levels, 53 dB L_den_ and 45 dB L_night_ for road traffic noise and 54 dB L_den_ and 44 dB L_night_ for railway noise, respectively [52], indicating slightly lower thresholds for health effects for aircraft noise in comparison to road traffic noise and railway noise.

Noise is hypothesised to influence cardiovascular health by causing physiological stress reactions in an individual, leading to increases in cardiovascular disease risk factors, such as high blood pressure, blood lipids and blood glucose, which overtime can manifest as cardiovascular diseases such as hypertension, ischaemic heart disease and stroke [53]. Noise and noise annoyance are also hypothesised to increase stress hormones, such as catecholamines (e.g., adrenaline and noradrenaline) and cortisol, with prolonged activation of these hormones leading to the development of depression and anxiety disorders [54]. Sleep disturbance itself has a range of health impacts, including impairing mood, impairing cognitive performance and increasing sleepiness the next day [45]. Sleep disturbance could also contribute to cardiovascular and metabolic disease by influencing glucose metabolism, appetite regulation, immune responses and dysfunction of blood vessels [45].

Annoyance is likely to be one of the key health outcomes to assess in relation to new aviation technologies: it is an established measure of community response and standardised questions for its measurement in socio-acoustic surveys are available [42]. Annoyance is commonly used by policy makers and communities alike to measure the quality of life impact of environmental noise exposure around airports. Communities are aware of the health effects of aircraft noise.

Alongside annoyance responses, depending upon the operating restrictions, effects on sleep and children’s learning should be relevant and immediate considerations in communities exposed to UAV noise. Effects on cardiovascular and metabolic ill-health, if present, would take several years to manifest; so, it may be more challenging to study in the initial stages of developing UAV operations and systems. This is not to suggest that planning for UAVs should ignore cardiovascular and metabolic health; however, it may be necessary to take a precautionary approach and to rely on the exposure–response functions of the aircraft noise effects on these outcomes to estimate the effect of UAVs on public health, in the short term. It may also be possible to build knowledge in the laboratory about the short-term biological response to UAV noise [55], but there will be inherent risks in extrapolating this knowledge to long-term biological responses in the community.

ERFs are the main tool for assessing community noise impact caused by transportation noise. There are well-established ERFs for transportation noise and annoyance (e.g., [56]) that have been the basis upon which the recommended noise exposure levels for transportation have been set. Prior to wider adoption of UAVs, it is necessary to build knowledge about ERFs for different vehicle configurations, as well as for different operational contexts, such as day/night and urban/suburban/rural settings. Initially, building this knowledge will only be possible under laboratory conditions. The information gathered may be used to inform certification, airspace management, infrastructure design and planning policy; later on, it should be verified in the field when communities are exposed to the operation of these aerial vehicles. The expectation is that this knowledge built in the laboratory would aid industry to introduce UAVs without a significant increase in community annoyance.

The need to establish ERFs for UAVs will also be influenced by the development of knowledge about the appropriate noise metrics. For cases where the metrics match those typically used for transportation noise assessment, e.g., energy-averaged metrics, such as L_Aeq_,_16h_, L_night_ and L_den_, it will be possible to compare the annoyance responses for UAVs to the annoyance of other sources, such as aircraft noise generated by civil aviation. This will be important knowledge to build, whilst appreciating that it is likely that other ERFs for new noise metrics (optimised for UAVs) may also need to be established for these new aerial vehicles.

Several studies published in recent years have suggested an increase in aircraft noise annoyance around major airports in Europe [57,58,59]. In these studies, communities report greater levels of being highly annoyed at the same sound level (measured in energy-averaged metrics) than in previous decades. A number of reasons may explain these findings. Communities have become more organised in their response to aircraft noise, given the greater knowledge about the health effects of noise and the increased use of social media to organise responses. However, this increase in reported annoyance could also be linked to an increase in the number of events rather than the noise level per se, as many airports have seen a slight reduction in exposure, as assessed by energy-averaged metrics, but a considerable increase in the number of events. The “number of events” metric has become important for communities exposed to aircraft noise, who often feel that energy-averaged metrics do not adequately reflect their experiences of noise exposure. Communities are more organised to oppose increases in aircraft noise exposure or becoming newly overflown than perhaps they were in the past, and for this reason the introduction of a new source of noise may lead to tension.

Civil aviation in the developed world had a mostly reactive approach to noise annoyance in local communities, with the sector being fairly slow to engage with local stakeholders and communities. Too often, it has been the case that only once local communities have organised themselves to respond that airports have engaged and consulted. UAV operators must avoid falling into the same trap. In some jurisdictions there has also been increasing policy need for airports to consult with local communities [60]. Whilst tracking human responses from the laboratory to the field may pose some methodological issues (discussed below), the opportunity to develop knowledge in the laboratory should not be passed up.

The operation of UAV infrastructure will also need to bear in mind the public health impact of newly exposing communities to noise. In civil aviation, studies of change in aircraft noise exposure, including studies of newly overflown communities, have found that there is an excess response in relation to the change in noise exposure, both for decreased and for increased aircraft noise exposures [61,62,63,64,65]. When noise exposure is changed, subjective reactions may not change in the way that would be predicted by steady-state ERFs; i.e., the annoyance response is likely to be slightly higher than that predicted by the noise exposure. This effect has been found for both aircraft noise and road traffic noise [62], suggesting that it could also be relevant for UAV noise. This knowledge is likely to be relevant for the introduction of new aerial vehicles, where communities will be newly overflown. Whilst UAV noise is likely to be at lower levels than typically seen for aircraft noise and road traffic noise exposure, the potential impact of the introduction of a new noise exposure, even if at levels of <55 dBA (L_A,max_ at 400 ft, see Section 2), could result in a negative community response, as well as health effects. The recent World Health Organization Environmental Noise Guidelines for the European Region have highlighted the low noise exposure thresholds, albeit in L_den_ and L_night_ metrics, for noise effects on health. We should not assume that there is a high noise threshold for effects. We have learnt in the past couple of decades, from studying the effects of aircraft noise and road traffic noise on a range of health outcomes, including annoyance, sleep and cardiovascular health, that the effects start at a much lower exposure than was previously assumed [47,52]. However, knowledge about the health effects for metrics that may be relevant for UAVs, such as L_A,max_ and the number of events, remains to be established for most environmental noise sources. A recent national survey established an ERF between the number of events (number of events ≥65 dB L_A,max_) and annoyance, indicating a sharp increase in annoyance responses between 50–99 events and 100–199 events per day [66]. The Swiss SiRENE study found that the ERF between aircraft noise exposure (L_den_) and annoyance varied slightly with intermittency, suggesting that annoyance was slightly higher for continuous compared to intermittent aircraft noise [32].

Non-acoustic factors have an important influence on annoyance responses to environmental noise [40], and therefore should also be considered in relation to UAV noise exposure. A variety of individual, social, situational and environmental factors have been found to moderate the relationship between transportation noise and annoyance or disturbance responses [40]. Some examples are (1) individual factors, such as noise sensitivity, coping capacity and attitude towards the source, such as fear or acceptance, age, gender and socioeconomic status, influence annoyance responses (and also vary considerably between people); (2) social factors, e.g., the social value placed on the noise source by a community or society; (3) situational factors, related to the immediate context, such as night-noise causing an awakening or an increase in flights over the years; and (4) environmental factors, such as access to quiet or recreational areas or the presence of a quiet side of the home. Such factors have been shown to have a powerful influence on annoyance responses [40,41,66], and knowledge about these factors, particularly in relation to individual, social and situational factors, will need to be built in relation to UAVs.

A recent report [67] points out that safety, privacy and security, in addition to noise and visual pollution, are the main concerns regarding the use of UAVs. Sparrow et al. [44] suggested that there is some evidence that the public may be concerned with these new noise sources, and that further research is deemed to understand human responses to UAV noise. As observed for aircraft noise, attitudes to the source and the social value placed on the noise source are likely to be key factors in the community response to UAVs.

### 4.3. Managing Noise

Aircraft noise management around airports is most usually based on the “Balanced Approach” developed by the International Civil Aviation Organization (ICAO) [68]. The Balanced Approach puts forward four principles with the aim of limiting the number of people exposed to aircraft noise: (1) Reduction of noise at the source; (2) land-use planning and management; (3) noise abatement operational procedures; and (4) operating restrictions on aircraft. These principles may be considered through the development and deployment of UAVs. For instance, there exist opportunities to reduce the noise at source through changes in the design space [69], as well as through consideration of the infrastructure and operational procedures. As an emerging technology, there is a unique opportunity to review and update the Balanced Approach for UAVs through incorporating knowledge about human response in the vehicle design stage [70].

Regarding the other three principles, operating restrictions are seen as the last option to be adopted, only to be implemented if the first three principles have been exhausted. Important caveats should be considered for the land-use planning and noise abatement procedures. As said above (see Section 3), the noise management of civil aviation (i.e., Balanced Approach) is focused on takeoff and landing operations taking place in the vicinity of an airport; however, UAV noise management should consider both takeoff and landing operations, but also on cruise operations (as these vehicles will flyover close to a population). Industry is understanding this, and they are planning the operations of their vehicles so that they do not cause a significant change in the typical ambient sound at a particular time of the day. As an example, Uber Elevate is planning their routes and skyport locations, so the operations of their vehicles do not increase the ambient equivalent continuous sound level (L_Aeq_) by more than 1.5 dBA [71].

A soundscape approach can provide solutions for an efficient planning of UAV operating procedures. For instance, Torija et al. [7] investigated the effects of a UAV operation (i.e., hovering) on the perception of different urban soundscapes. These authors found that, even with the same UAV operation at the same sound level, the increase in reported annoyance with the UAV hovering was significantly higher in soundscapes with reduced road traffic noise than in soundscapes highly impacted by road traffic. The noise annoyance with the UAV hovering was reported up to 6.4 times higher than without the UAV hovering, in locations with low road traffic volumes. These results seem to suggest that the operation of UAVs in corridors along busy roads might aid in the mitigation of the overall community noise impact caused by these aerial vehicles. However, these results also suggest that “as traffic varies over the day there will be times with low traffic volume when local residents experience the loudness and annoyance of drones much more” [72]. This supports examining metrics for UAVs for shorter-time periods to assess their effects at different times of the day.

Consideration of how the ambient, existing soundscape can influence responses to UAVs raises interesting issues regarding how a UAV will interact with the existing environment, but also raises the important issue of equity. We already know that, in many countries, exposure to environmental noise such as road traffic and aircraft noise is disproportionately distributed in the population, with higher noise exposures in groups with a lower socioeconomic position [73,74]. There is a risk that UAV operation could be disproportionately cited and routed inequitably and further add to the environmental burden already experienced in relation to noise exposure. It should be remembered that even small increases in noise exposure could be significant in terms of health as well as human and community response if the existing ambient exposure is of a moderate level.

## 5. Recommendations for Research Requirements

Given the sheer scale of the task ahead, it is clear that an experimental approach to building knowledge will need to be large-scale and ambitious. There is a need to build knowledge urgently. There is already funding support available in some countries (e.g., the Future Flight Initiative in the UK; the Grand Challenge in the United States) for research in this field, but it is clear that there will need to be multiple initiatives and collaborations for knowledge to be built to inform the ongoing, rapid development of electric and autonomous flying technologies. NASA’s white paper [75] has summarised the research gaps and discussed recommendations to address the barriers associated with Urban Air Mobility noise. This white paper suggests the following areas of interest: (1) prediction tools and technologies for noise reduction; (2) ground and flight noise testing; (3) human response and metrics; and (4) regulation and policy.

Simulated auralisations (immersive sound recordings) and visualisations of UAVs presented in laboratory settings offer a well-controlled, fully calibrated immersive listening environment for laboratory studies and could be used to address the following challenges in relation to human response and public readiness:Develop knowledge about human responses to the sound produced by these new aerial vehicles, and understand public acceptance.To evaluate the sound emissions of vehicles to identify the appropriate metrics to describe the exposure. This information could also inform certification standards, assessment methods and policy. The development of metrics will need to be informed by human response to noise, as well as the ability of individuals to understand what the metric represents for communicating environmental impacts with communities.To create virtual reality sound demonstrations to demonstrate the new technologies in their context to communities.

Both qualitative and quantitative research methodologies will be needed to develop knowledge and there would be advantages from developing standardised methodologies for use in studies that can be used and adapted globally. Both methodologies will involve listening studies in laboratories, as well as large-scale virtual reality studies undertaking a holistic assessment of the sound and visual impact of vehicles for different types of communities. Ecological validity—that is, how well the knowledge developed in the laboratory corresponds with experience in the real-world—will also need to be assessed in real-life tests; this is likely to initially involve experimental opportunity samples as opposed to community samples, where responses to stimuli in the laboratory setting can be compared with responses to stimuli in a simulated real-life setting.

Explorative and developmental work on UAVs should be undertaken using qualitative approaches, such as interviews and focus groups. Qualitative research will be essential for developing knowledge about public readiness for these new technologies, as well as tracking public attitudes over time, including before and after introduction of UAVs.

In terms of quantitative studies, it will be important to develop a plan for knowledge development: to conduct studies in an ordered fashion; to enable knowledge to be built; to not overstate the findings of studies; and to acknowledge uncertainties. There will be a careful balance to be sought between having one eye on the bigger picture of how the research is building and thinking about the aims of each research study. Given the need to build knowledge quickly, it will be important to resist the temptation to manipulate too many variables at once in a laboratory setting. Initially, more will be learnt from keeping most of the factors of interest stable and manipulating one factor at a time. Then, we can slowly develop studies that manipulate the other factors in turn or in controlled conditions. Knowledge building will be needed to inform holistic knowledge and is tied to developing clear research questions. We should always use the results of the previous studies to inform the design of the next study, where relevant.

Table 2 shows a range of metrics that could be explored in these types of studies. This includes traditional metrics, such as sound quality metrics—loudness, sharpness, tonality, roughness, fluctuation strength and impulsiveness—as well as average metrics and psychoacoustic and soundscape descriptors.

The development of ERFs for UAV noise is a significant challenge. Without the deployment of UAVs at scale, and therefore the wider communities exposed to the noise of these aerial vehicles, it is highly unlikely that enough evidence can be gathered to derive robust ERFs. As described above, some laboratory studies have already compared the perception of UAV noise with the perception of the noise generated by civil aircraft [18] and road vehicles [8]. Although immersive sound scenes can be simulated in laboratory studies, it is uncertain whether the findings of these studies could be extrapolated to real scenarios. This question remains as previous evidence for conventional aircraft noise suggest an important contribution of several non-acoustic factors (see Section 4.2). A possible solution may be to apply the annoyance “bonus” or “penalty” for UAV noise found in the lab (by comparing UAVs with other conventional vehicles) to standard ERFs for transportation noise, and thus derive pseudo-ERFs for UAVs. However, these ERFs may still need to be validated with extensive field studies before they could be adopted for UAV noise regulation. Further research is needed to develop robust approaches to anticipate the community noise impact of UAV fleets.

## 6. Conclusions

This paper introduces and discusses the main challenges to and research gaps in understanding the human responses to UAVs. The paper outlines the main noise characteristics of these novel aerial vehicles and states the need for further research to understand the effect of UAV noise on public health and wellbeing: to develop metrics to assess the community noise impact of UAVs; to define acceptable levels for UAVs; inform best operational practices for drones with regard to noise profiles; and to innovate approaches to predict the long-term noise effects when UAVs operate at scale. Bridging these research gaps is crucial to appropriately tackle the noise issues associated with UAVs, and therefore protect the quality of life and health of exposed communities. Given the scale of the task ahead, it is clear that the approach to building knowledge will need to be large scale, ambitious and embrace different research methodologies, bridging the gap between engineering and human response to noise disciplines. The field of human response to noise exposure is typically undertaken by those with a public health and/or social science background and there will be a need to embrace these disciplines, alongside expertise in public acceptance and community engagement, to build this knowledge.

## Figures and Tables

**Figure 1 ijerph-18-00682-f001:**
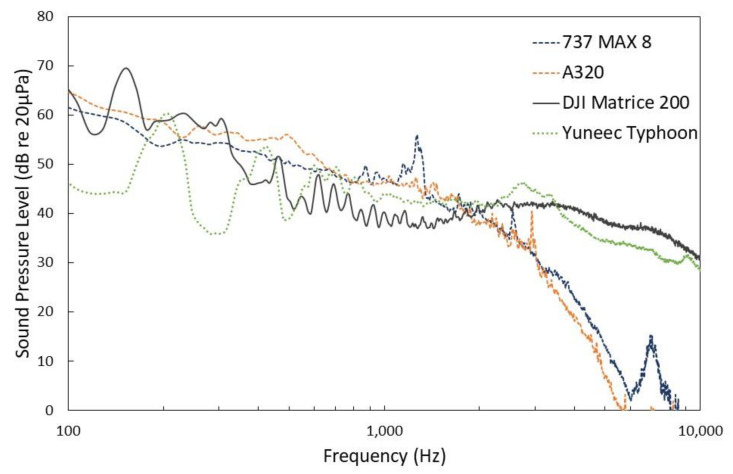
Frequency spectra of two conventional aircraft (Airbus A320 and Boeing 737-8MAX) and two multi-copter Unmanned Aerial Vehicles (DJI M200 and Yuneec Typhoon [9]). Frequency spectra normalised to 65 dB(A).

**Figure 2 ijerph-18-00682-f002:**
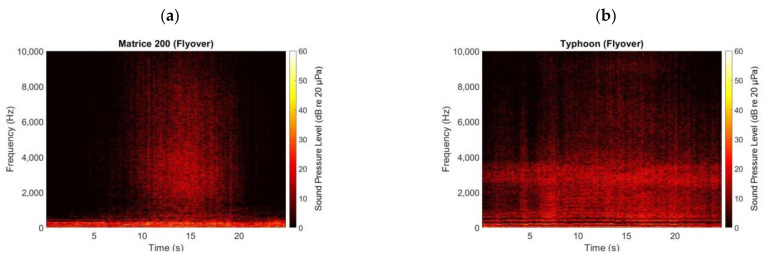
Spectrogram of the DJI M200 (**a**) and Yuneec Typhoon (**b**) UAVs flying over at ~50 m (~150 ft) altitude above the microphone [9].

**Figure 3 ijerph-18-00682-f003:**
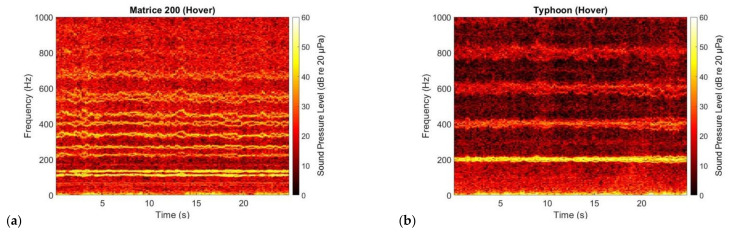
Spectrogram of the DJI M200 (**a**) and Yuneec Typhoon (**b**) UAVs hovering at ~1.2 m (~4 ft) altitude above the microphone [9].

**Table 1 ijerph-18-00682-t001:** L_A,max_ values at 10 km and 25 km from the Start-of-Roll (SOR) point for both an Airbus A320 and Boeing 737-8MAX.

Aircraft	Distance from SOR Point (km)	Height about Ground (ft) ^1^	L_A,max_ ^2^
A320	10	2400	73.4 dB(A) (SD = 4.3 dB(A))
737-8MAX	10	2200	71.9 dB(A) (SD = 3.0 dB(A))
A320	25	6500	60.2 dB(A) (SD = 4.6 dB(A))
737-8MAX	25	7500	55.6 dB(A) (SD = 3.3 dB(A))

^1^ Estimated using the A320 and B737-800 height profiles in [11]. ^2^ Average values of L_A,max_ for takeoff operations with the power setting ranging from 12,000 lbf to 22,500 lbf (Airbus A320), and from 10,000 lbf to 24,500 lbf (Boeing 737-8MAX).

**Table 2 ijerph-18-00682-t002:** Variables to be explored to build knowledge of the human responses to UAVs.

Sound Quality Metrics	Human Response Outcomes ^1^	Psychoacoustic Factors	Other Factors
Loudness	Annoyance ^2^	Pleasantness/Eventfulness	Visual impact
Sharpness	Audibility	Calmness/Vibrancy	Ambient noise
Tonality	Physiological responses		Community soundscape
Roughness	Sleep disturbance		Indoor versus outdoor perception
Fluctuation strength	Cognitive effects		Contextual/cultural/population variation
Impulsiveness	Perceived stress		Vehicle type variability
Sound Exposure Level (L_AE_)			Number of events
L_Aeq_, T			Time of day
L_A,max_			Altitude
			Attitudes to source/operators
			Perceptions of safety/trust

^1^ Plus additional factors identified through qualitative research. ^2^ ISO/TS 15666 methodology [42].

## Data Availability

Restrictions apply to the availability of the UAV audio recordings analysed in this paper. There audio recordings were obtained from John A. Volpe National Transportation Systems Center. Access to this data can be requested to David R. Read, Christopher Cutler and Juliet Page (John A. Volpe National Transportation Systems Center).

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
