# Peer review of "A Psychoacoustic Approach to Building Knowledge about Human Response to Noise of Unmanned Aerial Vehicles"

_ijerph, 2021, doi:10.3390/ijerph18020682_

Round 1

Reviewer 1 Report

The operation of UAVs will lead to noise exposure, which has the potential to become a significant public health issue. This paper describes the main acoustic and operational characteristics of UAVs. Gaps in literature and regulation on noise metrics and acceptable noise levels are identified and discussed. Based on reviewing human response to aircraft noise, methodological framework is proposed for building psychoacoustic knowledge to inform systems and operations development for a minor noise impact on communities. A Suggestion of  minor revision is below.

Substitute ‘electric Vertical Takeoff and Landing (eVTL) (eVTOL)’ for ‘electric Vertical Take Off and Landing’.

Why did frequency spectra be normalized to 65 dB(A) in Figure 1? It is better to give an actual frequency spectrum.

Reviewer 2 Report

The article discusses the noise of Unmanned Aerial Vehicles (UAVs), and its influence on humans. It reviews and discusses the topic that is new, since so far UAVs are not that common source of noise. It describes UAV noise characteristics, possible future influences UAV noise might have on humans, and aspects that should be taken into account while the UAV noise becomes more common. As such, it is a good introduction to the topic, well written, and nice to read. It is an interesting addition to the area of environmental noise research.

I have just minor comments listed below as well as few more general questions.

The noise described in the article is mainly from the UAV that is hovering or flying in one place. This noise is tonal, which is often more annoying for a human than a noise at similar sound level without this special characteristic. This is ok. This way of describing the noise is mainly due to how the noise is measured and its limitations are discussed in the paper. However, the drones fly and generally they fly from one place to another, therefore, their noise is normally not steady hovering, but it is a bypassing event for a person being in one place. This eventfulness is something that the authors have mentioned here and there. However, I would like it to be discussed more, since I think it is an important characteristic of this noise. In addition to eventfulness, these events will be unpredictable. These events can be the reason why sound at lower sound levels is considered annoying. It might be the Lmax and possibly the number of events that would better predict annoyance and other metrics measuring average sound level over time. These events might distract people away from what they are doing and take their attention to the event etc.

In lines 119-123 you write: ” For all these reasons, authors like Christian and Cabell [8] have suggested that the sound produced by drones does not resemble qualitatively the sound generated by contemporary civil aircraft.” In addition, you list other differences between UAV noise and aircraft noise: UAV noise is more tonal and has higher sound frequencies, the noise is not steady, but changes depending on weather conditions and control system to maintain vehicle stability, etc. Why do you then discuss the human response to UAV noise by comparing it with the effects of civil aircraft noise? Why is this a good comparison and not a comparison with other type of environmental noise? Please, justify why this environmental noise is a good and relevant comparison.

Minor comments:

Line 37

“Schäfer et al. [1] discussed how a transition to all-electric aircraft might significantly reduce the environmental of aviation.” --> Missing a word?

Line 42

“Elsayed and Mohamed [4] found that parcel delivery using UAVs produces lower CO2 emissions than ground delivery per parcel-km.” --> Is this the same if vehicles on the ground start being electric or on what this estimation was based on?

Lines 64-66

Furthermore, this paper describes a psychoacoustic approach to build knowledge about human response to UAV noise, to inform the design and operation of UAVs for a minor impact on quality of life and health.

  • Minor impact or possible?

Line 71

Is this (55.1 ±6.2 dB(A)) a mean and standard deviation? Write it open.

Lines 74-75

“According to the Aircraft Noise and Performance (ANP) Database [10], at 400 74 ft, the LA,max of a conventional Airbus A320 (CFM56-5A1 engine) and Boeing 737-8MAX 75 (CFM Leap1B-27 engine) is 95.6 ±2.9 dB(A)1 and 90.1 ±3.1 dB(A)2 respectively.”

  • Is the comparison sensible, since the noise from aircrafts is not from 400 m distance unless they are close to take-off or landing? I would like to know how often the noise from aircrafts would be from this distance and what the distance to aircraft and possibly noise close to human habitat normally is.

Line 136

) is missing.

Line 191-192

“Impulsiveness and tonality account for the perception of sudden changes in the sound level and the presence of tonal components respectively.” --> This might be clearer in separate sentences.

Lines 197-199

“For a single quadcopter (DJI Phantom 3) with different payload and for flyover operations, Torija and Li [16] found that tonality and loudness-sharpness interaction were the two main psychoacoustic factors determining the preference of the UAV sounds evaluated.”

  • Try to open this better. I do not understand what the result was. Which characteristics predicted preference?

Line 228

exposure-response relationships (ERFs) --> Why relationship is abbreviated with F? Would it be better to state here exposure response function (ERF)?

Line 281

Lnight --> Lnight

Line 314

“This effect has been found for both aircraft noise and road traffic noise” --> References would be nice here.

Line 405

UAM noise – You are using abbreviation UAM just twice in the paper – in the introduction and Chapter 6. Perhaps it is clearer to use just Urban Air Mobility? In addition, think whether SEL is a necessary abbreviation. Try to avoid unnecessary abbreviations that you are not using many times. Check the paper for these.

Table 1. You have cognitive load in the table. Learning effects of children is one possible effect, which I think is not described well by the term cognitive load. Consider revising this by changing cognitive load for example into cognitive effects or adding learning difficulties into the table.

Reviewer 3 Report

     At this time, with interest growing in the use of unmanned aerial vehicles, I believe it is essential that studies raise questions about the impact of these vehicles on the health and well-being of people who will be exposed, especially to the noise of these vehicles.  The authors, well versed in the research on the impacts of noise, including those from aircraft, on the health of residents exposed to such noises, have diligently explored the need for research on the measurements of sound levels to which individuals will be exposed when unmanned vehicles fly above them and the potential impacts of these sound levels on residents including short and long-term effects. 

     This paper is important in that it has "opened the door" to the kind of research that is imperative as we explore the future use of unmanned aerial vehicles.  As the authors so aptly noted, this research must be conducted by individuals knowledgeable about sound measurements as well as those more informed about the effects of noise on health.
